# Location-Aware Visual Question Generation with Lightweight Models

**Nicholas Collin Suwono**[1,2]   **Chih Yao Chen**[1]   **Tun Min Hung**[1]
**Ting-Hao 'Kenneth' Huang**[3]   **I-Bin Liao**[4]   **Yung-Hui Li**[4]   **Lun-Wei Ku**[1]
**Shao-Hua Sun**[2]
Institute of Information Science, Academia Sinica[1]   National Taiwan University[2]
Pennsylvania State University[3]   Hon Hai Research Institute[4]
r10946021@ntu.edu.tw   cyaochen@cs.unc.edu   hungtungming@gmail.com
txh710@psu.edu   yunghui.li@foxconn.com   ibin.liao@foxconn.com
lwku@iis.sinica.edu   shaohuas@ntu.edu.tw

## Abstract

This work introduces a novel task, location-aware visual question generation (LocaVQG), which aims to generate engaging questions from data relevant to a particular geographical location. Specifically, we represent such location-aware information with surrounding images and a GPS coordinate. To tackle this task, we present a dataset generation pipeline that leverages GPT-4 to produce diverse and sophisticated questions. Then, we aim to learn a lightweight model that can address the LocaVQG task and fit on an edge device, such as a mobile phone. To this end, we propose a method which can reliably generate engaging questions from location-aware information. Our proposed method outperforms baselines regarding human evaluation (*e.g.*, engagement, grounding, coherence) and automatic evaluation metrics (*e.g.*, BERTScore, ROUGE-2). Moreover, we conduct extensive ablation studies to justify our proposed techniques for generating the dataset and solving the task.

## 1   Introduction

Driving is an integral part of our daily routines, playing a significant role in our lives. Whether commuting to work, running errands, or embarking on exciting adventures, we heavily rely on automobiles to get us from one place to another. Despite its undeniable convenience, driving requires constant focus on the road, the need to remain alert, and the mental strain of navigating through traffic. Hence, staying behind the wheel after long working hours or during a long-distance trip can give rise to hazardous circumstances. To combat this, passengers often engage in conversation to keep the driver awake and attentive.

Can we develop a system running on a lightweight device that automatically engages the

---

Project page can be found at https://github.com/AcademiaSinicaNLPLab/LocaVQG

Figure 1: **Location-aware Visual Question Generation (LocaVQG)** involves generating engaging questions from a specified location, represented by a GPS coordinate of a vehicle and a set of street view images captured by on-car cameras.

driver in a conversation? While initiating a conversation with general questions may not interest the driver, delving into driver-specific inquiries raises privacy concerns since it requires personal information. Our key insight is to engage the driver in a conversation by posing questions based on the location-aware information, composed of both the geographical coordinate of the car and surrounding visual perception represented by pictures captured by on-car cameras. Such rich location-aware information allows for producing diverse and relevant questions, enabling a system to initiate an engaging conversation.

In this work, we introduce a novel task, Location-aware Visual Question Generation (LocaVQG), which aims to produce engaging questions from a GPS coordinate of a vehicle and a set of street-view images captured by on-car cameras, as illustrated in Figure 1. We make the first attempt to tackle this task by developing a data generation pipeline that can create a dataset containing high-quality samples for the LocaVQG task. To this end, we leverage the recent advances in large language models (LLMs) (Liu et al., 2023; Touvron et al., 2023).

Specifically, we collect data from Google Street View and design a prompt according to the address obtained by reverse geocoding the GPS coordinate and the captions of street-view images provided by an off-the-shelf image captioning model. While LLMs can generate questions relevant to the provided location-aware information, the produced questions may not always be engaging. Therefore, we further propose to train an engaging question classifier that can filter out non-engaging questions. Our proposed dataset generation pipeline is illustrated in Figure 2.

We present a method, FDT5, that can learn a lightweight model and reliably address the LocaVQG task. We compare our proposed method to various small and mid-size language models learning from the generated dataset. The experimental results demonstrate that our proposed FDT5 outperforms the baselines regarding human evaluation (*e.g.*, engagement, coherence, grounding) and automatic evaluation metrics, *e.g.*, BERTScore (Zhang et al., 2019), ROUGE-2 (Lin, 2004). Our FDT5 with only 15M parameters achieves competitive performance even compared to a large language model (*i.e.*, GPT-4). This highlights the effectiveness of the proposed dataset generation pipeline as well as the proposed training techniques.

The main contributions of this work are threefold as follows:

- **Task.** We propose Location-aware Visual Question Generation (LocaVQG), a novel task that aims to produce engaging questions from a GPS coordinate of a vehicle and a set of street-view images captured by on-car cameras. This will lead to the development of more intelligent in-car assistant systems.

- **Dataset.** To address LocaVQG, we introduce a dataset generation pipeline that can produce diverse and engaging questions from a specified location by leveraging pre-trained LLMs.

- **Method.** We present a method FDT5 that outperforms all the lightweight baselines regarding human evaluation (*e.g.*, engagement, coherence, grounding) and automatic evaluation metrics (*e.g.*, BERTScore, ROUGE-2).

## 2 Related Works

**Self-driving cars.** Despite the recent advances in developing self-driving cars (Parekh et al., 2022),

most current commercialized autonomous vehicles are categorized as SAE (the Society of Automotive Engineers) (International, 2018) Level 2 (*e.g.*, Tesla, Hyundai, Kia) or Level 3 (*e.g.*, Mercedes). When driving an SAE Level 2 vehicle, the driver must always hold the steering wheel. With an SAE Level 3 vehicle, the driver must still be ready to take control of the vehicle at all times when prompted by the vehicle. That being said, driving in modern days still requires the driver's attention and therefore can be assisted with the task and the system proposed in this work.

**In-car intelligent assistant system.** Developing in-car intelligent assistant systems, such as a voice assistant (Lin et al., 2018; Braun et al., 2019), is an emerging research area. Large et al. (2017) discovered that engaging drivers in a conversation could effectively reduce driver fatigue. In contrast, this work focuses on raising drivers' attention by formulating a task and devising a system to produce location-aware engaging questions.

**Visual question generation (VQG).** VQG concerns generating questions from visual inputs (Mostafazadeh et al., 2016). Compared to this work, recent works (Lu et al., 2021; Yeh et al., 2022) that explore VQG do not leverage geographical information (*e.g.*, GPS). On the other hand, Zhang and Choi (2021) presented a dataset with questions respecting geographical and temporal contexts; yet, it does not utilize visual inputs. In contrast, the task and the dataset proposed in this work leverage images captured by on-car cameras.

**Large language models (LLMs).** Recent advances in LLMs (Liu et al., 2023; Touvron et al., 2023) have led to promising results in various domains (Wei et al., 2022; Zhang et al., 2023). However, these gigantic LLMs with billions of parameters (Liu et al., 2023; Touvron et al., 2023) cannot be deployed on lightweight devices and therefore are not ideal for in-car intelligent assistant systems. This work aims to develop lightweight models that can run on edge devices like mobile phones.

**Lightweight language models.** Existing mobile-friendly language models (Sun et al., 2020; Mehta and Rastegari, 2021) struggle at language generation tasks. This work aims to devise lightweight models that can address VQG tasks and achieve competitive results even compared to LLMs.

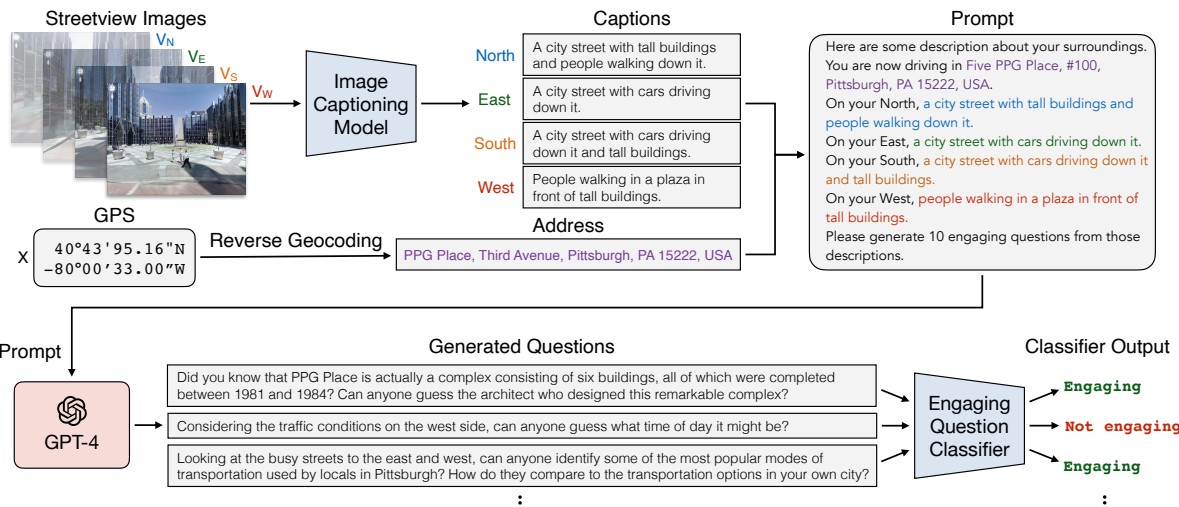

Figure 2: **Data Generation Pipeline**. This pipeline produces questions from a given LocaVQG task tuple consisting of four street view images $V_N, V_E, V_S, V_W$ and a GPS coordinate $X$. We caption each image using an image captioning model and infer the address by reverse geocoding the GPS coordinate. Then, we construct a prompt that describes in detail the location-aware information and leverage GPT-4 to generate questions. We further employ an engaging question classifier to filter out non-engaging questions. Finally, the remained questions are included in the dataset with the given LocaVQG task tuple.

## 3 LocaVQG: Location-aware Visual Question Generation

We introduce a novel task, Location-aware Visual Question Generation (LocaVQG). This section formally defines this task and describes how we collect data to construct the LocaVQG task tuples.

### 3.1 Location-aware Information

Location-aware Information includes data or content specifically relevant to, or influenced by a particular geographical location. With such information, applications can offer location-specific recommendations, directions, local weather updates, nearby points of interest, targeted advertisements, etc. Since our goal is to produce engaging questions with an in-car device based on location-aware information, we limit it to the information that is easily accessible even without the internet. Specifically, we consider the surrounding visual perception and the geographical coordinate.

### 3.2 LocaVQG Task Tuple

To collect the surrounding visual perception and the geographical coordinate of diverse locations, we propose to leverage Google Street View Dataset (Zamir and Shah, 2014). The dataset contains 10,343 coordinates, and each coordinate comes with 5 corresponding directional images (North, East, South, West, and Upper/Sky view).

To ensure the location diversity, we select 3,759 coordinates with their 4 directional images, excluding the upper/sky view, which is usually not observable by the driver. We denote the geographical coordinate of each location as $X$ and its surrounding images as $V_d$ with $d = [N, E, S, W]$, standing for each direction. We define our LocaVQG task tuple $T$ as: $T = [V_N, V_E, V_S, V_W, X]$. Given a LocaVQG task tuple $T$, our goal is to produce an engaging question $Q$ with a model $f$: $f(T) = Q$.

## 4 Generating LocaVQG Dataset

Our goal is to train lightweight models to address the LocaVQG task. Therefore, we aim to "label" the task tuples described in Section 3. Annotating the task tuples with engaging questions requires creativity and location-specific domain knowledge, which can be challenging for human annotators. In this work, we propose automatically generating questions from LocaVQG task tuples by leveraging the recent development of LLMs. An overview of the proposed dataset generation pipeline is depicted in Figure 2.

### 4.1 Prompting GPT-4

This section describes how we utilize GPT-4 (OpenAI, 2023) to produce questions from LocaVQG task tuples by processing task tuples and designing LocaVQG prompts.

| Engaging Questions |
| --- |

The city of Pittsburgh is known for its numerous bridges. How many bridges do you think are in the city, and why do you think there are so many?

What types of events or festivals might take place in this park throughout the year?

As we look towards the south, can you guess the purpose of this brick building with cars parked in front? Perhaps an office building, a restaurant, or something else?

| Non-Engaging Questions |
| --- |

Speaking of the hospital, does anyone know the range of medical services provided at Prince George's Hospital?

What are some ways that city planners might improve traffic flow at busy intersections?

Noticing the mixture of architectural styles, can you guess which era had the most significant influence on the city's development?

Table 1: **GPT-Generated Questions.** We provide examples of GPT-4 generated questions that are classified as engaging and non-engaging by the engaging question classifier. Answering these non-engaging questions often requires specific domain knowledge and therefore may interest only limited audience. blue-colored text indicate visual cues, red-colored text indicate directional cues, teal-colored text indicate location-specific information.

**Street view images → captions.** While GPT-4 is a multimodal model, its feature of taking image inputs is not yet publicly accessible as of May 2023. Hence, to inform GPT-4 with the street view images, we caption street view images using an off-the-shelf image captioning model (Wang et al., 2022).

**GPS coordinate → address.** To leverage the GPS coordinate, we reverse geocode it using Google's Reverse Geocoding API (Google), translating the coordinate into a street address. We found that with the decoded street address, GPT-4 can often infer nearby famous landmarks, or information, and generate relevant questions.

**Constructing prompts.** We aim to prompt GPT-4 with the processed location-aware input and produce engaging questions. We first design a **system prompt** that infuses GPT-4 with a tour guide role, enforcing it to engage users. Then, we design a **chat prompt** that encapsulates processed location-aware information and requires GPT-4 to generate engaging questions. The two prompts are presented as follows.

- **System prompt**: *You are a tour guide and you are driving in a car with your tourists. You want to engage with them with any kind of information you have around you.*

- **Chat prompt**: *Here are some descriptions of your surroundings You are currently driving on [Street Address]. On your North, [Image Caption]. On your East, [Image Caption]. On your South, [Image Caption]. On your West, [Image Caption]. Based on those descriptions, please ask 10 engaging questions.*

## 4.2 Filtering GPT-Generated Questions

While GPT-4 can generate numerous diverse questions from our designed prompts, we empirically find that some generated questions are not particularly engaging (*e.g.*, requiring domain knowledge), as shown in Table 1. To combat this, we propose to learn a BERT-based (Devlin et al., 2019) engaging question classifier to filter out non-engaging questions. We construct the training data for this classifier with non-engaging questions from SQuaD (Rajpurkar et al., 2016) and engaging questions from MVQG (Yeh et al., 2022). The key insight is that SQuaD questions are made for question-answering tasks, thus, solely revolves around a passage, while MVQG questions are collected with engagement in mind.

With this trained engaging question classifier, for each LocaVQG task tuple $T$, we filter out non-engaging questions generated by GPT-4, and the remained questions are included in the dataset as the "labels" for this task tuple.

## 4.3 Dataset Statistics

Applying the procedures described in Section 4.1 and Section 4.2 results in a dataset with 3759 task tuples and 35K questions. The basic statistics of the dataset are described in Table 2.

| | |
| --- | --- |
| # of LocaVQG Task Tuples | 3759 |
| - # of Task Tuples from Pittsburgh | 919 |
| - # of Task Tuples from Orlando | 611 |
| - # of Task Tuples from New York | 2217 |
| # of Questions After Filtering | 35551 |
| Average Sentence Length | 16.6 |
| Average Question Length | 30.8 |

Table 2: **Dataset Statistics.** We present the statistics of our location-aware visual question generation dataset

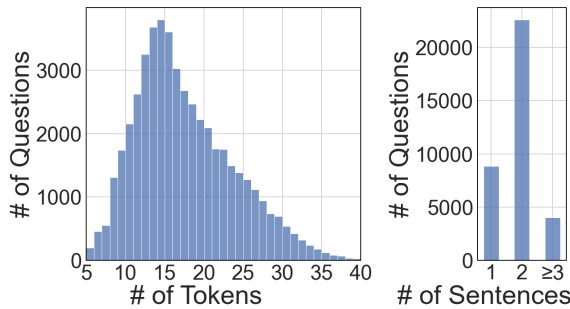

Figure 3: **Question Length.** We present the question length statistics in # and tokens and # of sentences.

### 4.3.1 Question Length

We present the histograms of question lengths in terms of # of tokens and # of sentences in Figure 3.

### 4.3.2 Frequent Trigrams and Words

We present the top 15 frequent trigrams in the dataset in Table 3. The questions often start by trying to intrigue the respondent (*e.g.*, Did you know, What do you). Also, open-ended questions (*e.g.*, *have you noticed*, *have you ever*) appear quite frequently. Most frequent words, presented in Table 4, require or lead the attention of the respondent (*e.g.*, *considering*, *looking*, *notice*).

| | | |
|---|---|---|
| Did you know | What do you | Can you spot |
| Can you guess | As we drive | How do you |
| Have any of | Have you noticed | Can anyone guess |
| Have you ever | Can you identify | What are your |
| Are you familiar | What are some | As we continue |

Table 3: **Top 15 Frequent Trigram of Questions.** The frequent trigrams appearing in the dataset show that the questions aim to intrigue or engage the respondent.

| | | | | |
|---|---|---|---|---|
| Speaking | Considering | Based | Looking | Notice |
| Since | Perhaps | Observing | Residential | New |
| Look | Pittsburgh | Given | Orlando | See |
| Let | Turning | Judging | Noticing | Besides |

Table 4: **Top 20 Frequent Words.** This table presents 20 frequent words appearing in the dataset aside from commonly used words, such as "*have*", "*can*".

### 4.3.3 Question Quality

We compare our proposed dataset to an engaging question dataset, the MVQG dataset (Yeh et al., 2022), regarding the following criteria and report the results in Table 5.

- **Vocabulary Size (Vocab Size)** measures the number of distinct words in a dataset.

- **Average Sentence Length (Avg Sent. Length)** computes the average length of sentences across the whole dataset, representing how rich and detailed a dataset is.

- **Syntactic Complexity** calculates the degree of variation, sophistication, and elaboration of the questions in a dataset (Ferraro et al., 2015). We report the mean of **Yngve Score** normalized by the sentence length.

- **Percentage of Abstract Terms (% Abstract Term)** computes the ratio of visual and non-visual concepts covered by a dataset, based on the abstract terms defined by Vanderwende et al. (2015).

- **Average Term Depth** is calculated based on WordNet, where noun words with a lower depth indicate higher-level concepts (Lu et al., 2021).

Compared to the MVQG dataset, the results show that our proposed LocaVQG dataset contains significantly more diverse and sophisticated questions. In fact, the questions included in the MVQG dataset are collected from human annotators. This highlights the effectiveness of generating questions by leveraging the recent advances in LLMs (*e.g.*, GPT-4), as proposed in this work. Further evaluations on the generated questions can be found in Section D and Section E.

| Criteria | MVQG | LocaVQG (Ours) |
|---|---|---|
| Vocabulary Size ↑ | 608 | **3046** |
| Average Sentence Length ↑ | 12.341 | **17.168** |
| Yngve Score ↑ | 2.271 | **3.761** |
| % Abstract Terms ↑ | 0.127 | **0.167** |
| Average Term Depth ↓ | 7.906 | **7.259** |

Table 5: **Question Quality Comparison with MVQG.**

## 5 Learning and Evaluating Lightweight Models

We aim to train and evaluate lightweight models learning from the proposed LocaVQG dataset.

### 5.1 Baselines

We compare our method to the following baselines. **Text-To-Text Transfer Transformer (T5).** We experiment with a family of T5 pre-trained language models (Raffel et al., 2020; Tay et al., 2021), which includes T5-Large (770M), T5-Base (220M),

| Model | #Parameters | Engagement | Naturalness | Coherence | Common Sense | Grounding | Overall |
|---|---|---|---|---|---|---|---|
| MVQG-VL-T5 | 254M | 3.84 | 3.64 | 3.65 | 3.81 | 3.84 | 3.76 |
| MVQG-VL-T5$_{fine-tuned}$ | 254M | 3.96 | 3.82 | 3.82 | 3.99 | 3.66 | 3.85 |
| T5-Large | 770M | 3.92 | 3.81 | 3.78 | 4.03 | 3.83 | 3.87 |
| T5-Base | 220M | 3.92 | 3.81 | 3.73 | 3.97 | 3.78 | 3.84 |
| T5-Tiny | **15.6M** | 3.96 | 3.79 | 3.67 | 4.01 | 3.81 | 3.85 |
| FDT5 (Ours) | **15.6M** | **4.03** | **3.83** | **3.96** | **4.05** | **4.03** | **3.98** |
| GPT-4 | 1T* | 4.12 | 3.99 | 4.01 | 4.05 | 4.01 | 4.04 |
| Human Annotator | - | 4.06 | 3.87 | 3.90 | 4.06 | 3.88 | 3.95 |

Table 6: **Human Evaluations.** Each question is rated by three AMT workers. Among all the light-weight models, our proposed FDT5 achieves the best overall performance and has the fewest parameters. Note that while the exact number of parameters GPT-4 is not revealed, many believe it is at least 6 times larger than GPT-3 (Brown et al., 2020) (175B).

and T5-Tiny (15.6M). We fine-tune the pre-trained T5 models on our LocaVQG dataset. Specifically, for each LocaVQG task, the models learn to map the prompt presented in Section 4.1 to one of the ground truth questions generated by GPT-4.

**MVQG-VL-T5.** Cho et al. (2021) introduced Vision-and-Language T5 (VLT5) for vision-and-language tasks. Yeh et al. (2022) adapted it for generating questions from a set of images. We adopt this method, dubbed MVQG-VL-T5, and fine-tune the pre-trained model on our LocaVQG dataset. The input of MVQG-VL-T5 consists of 4 street view images and the street address.

More details can be found in Section F.

## 5.2 Our Approach

We propose Filtered Distilled T5-Tiny (FDT5).
**Distillation.** While T5-Tiny has the fewest of parameters and can fit on mobile phones, its capacity might be limited to a complex task like LocaVQG. Therefore, we propose to learn a T5-Tiny model by distilling a learned T5-Large model. Inspired by Chen et al. (2020), during training, we utilize both the questions generated by GPT-4 from the dataset and the questions generated by the T5-Large model, resulting in the objective:

$$\mathcal{L}(\theta) = \alpha \cdot \mathcal{L}_{hard}(\theta) + (1 - \alpha) \cdot \mathcal{L}_{soft}(\theta), \quad (1)$$

where $\alpha$ balances the relative importance of learning from each loss and $\theta$ parameterizes the model. The hard-label loss $\mathcal{L}_{hard}$ (ground truth target) optimizes cross-entropy, while the soft-label loss $\mathcal{L}_{soft}$ (teacher model) optimizes KL Divergence.
**Filtering.** To further improve the engagingness of the questions produced by our method, we propose to utilize the engaging questions classifier described in Section 4.2 to filter out non-engaging

questions. Specifically, given a LocaVQG task, our method keeps generating questions until accepted (*i.e.*, classified as "engaging") by the classifier.

Our proposed method FDT5 combines the two techniques described above.

## 5.3 Human Evaluation

We provide human evaluation of the questions generated by all the methods.

### 5.3.1 Evaluation Metrics

We randomly sampled 100 LocaVQG task tuples and the questions produced by all the models. Each question is evaluated by three Amazon Mechanical Turk (AMT) workers according to the following metrics. We adopt a 5-point Likert scale for all the evaluation metrics.

- **Engagement**: You find the question engaging and you would want to answer the question.

- **Naturalness**: It is natural to ask this question given the information you have.

- **Coherence**: The question is coherent with the information you have.

- **Common Sense**: It makes sense to ask these questions given the information you have.

- **Grounding**: The question asked about things related to the information you have.

We also provide the evaluation of the questions generated by GPT-4, which can be considered as an upper bound as GPT-4 has an unparalleled number of parameters compared to the lightweight models. Furthermore, to compare the performance of these LMs against humans, we crowdsource and collect 100 questions on AMT based on the same set of

| Model | Engagement | Naturalness | Coherence | Common Sense | Grounding | Overall |
|---|---|---|---|---|---|---|
| Filtered Dataset (Ours) | **3.92** | **3.81** | 3.73 | **3.97** | 3.78 | **3.84** |
| Unfiltered Dataset | 3.89 | 3.76 | **3.78** | 3.85 | 3.78 | 3.81 |

Table 8: **Engaging Question Classifier for Dataset Generation**. Employing the engaging question classifier in the dataset generation process to filter out unengaging questions improves the quality of generated questions.

| Model | Engagement | Naturalness | Coherence | Common Sense | Grounding | Overall |
|---|---|---|---|---|---|---|
| Filtered Inference (Ours) | **4.03** | **3.83** | **3.96** | **4.05** | **4.03** | **3.98** |
| Unfiltered Inference | 3.96 | 3.78 | 3.82 | 4.04 | 3.79 | 3.88 |

Table 9: **Engaging Question Classifier for Inference**. Our proposed FDT5 employs the classifier during inference to filter out unengaging questions. Excluding the filtering phase results in significantly worse performance.

LocaVQG task tuples. More details on AMT can be found in Section G.

### 5.3.2 Results

The human evaluations are presented in Table 6.
**FDT5 outperforms all the lightweight models.** Our proposed method FDT5 achieves the best overall score with the fewest parameters. This justifies the effectiveness of our adopted distillation scheme. Furthermore, an average score of 3.98 indicates that our model can reliably generate satisfactory questions from location-aware information.
**MVQG-VL-T5.** MVQG-VL-T5, without learning from our dataset, achieves the worst performance, demonstrating the importance of constructing and learning from a dataset dedicated to the LocaVQG task. Alternatively, the MVQG-VL-T5 model fine-tuned on our dataset (MVQG-VL-T5$_{fine-tuned}$) struggles at grounding, aligning with the findings discussed in (Yeh et al., 2022).
**GPT-4 asks better questions than humans.** The questions produced by GPT-4 are preferred by the workers compared to those provided by human annotators on all the metrics, except for common sense. This justifies our proposed dataset generation pipeline, which collects questions from GPT-4 instead of humans.

### 5.4 Automatic Evaluation Metrics

| Model | BLEU-4 | ROUGE-2 | BERTScore | BLEURT |
|---|---|---|---|---|
| VLT5 | 0.2712 | 0.0342 | 0.5093 | -0.7208 |
| T5-Large | **0.2756** | 0.0380 | 0.5165 | -0.7336 |
| T5-Base | 0.2746 | 0.0388 | 0.5163 | -0.7305 |
| T5-Tiny | 0.2635 | 0.0371 | 0.5164 | -0.7419 |
| FDT5 (Ours) | 0.2661 | **0.0393** | **0.5190** | **-0.7073** |

Table 7: **Automatic Evaluation.** Our proposed FDT5 achieves the best performance on 3 out of 4 metrics (*i.e.*, ROUGE-2, BERTScore, and BLEURT).

We further evaluate the questions generated by all the models with some automatic evaluation metrics. To compare two questions based on exact wording, we are using BLEU-4 (Papineni et al., 2002), ROUGE-2 (Lin, 2004). To compare the questions based on semantic similarity, we are using ML-Based evaluation: BERTScore (Zhang et al., 2019), and BLEURT (Sellam et al., 2020). The results are presented in Table 7. Our proposed method FDT5 achieves the best performance on ROUGE-2, BERTScore, and BLEURT. T5-Large outperforms others in terms of BLEU-4. This verifies the effectiveness of the filtering and distillation technique employed in FDT5.

### 5.5 Ablation Study

We conduct extensive ablation studies to investigate the effectiveness of employing the filtering classifier (Section 5.5.1), the impact of incorporating GPS coordinates into the question generation process (Section 5.5.2), and the effect of varying dataset sizes (Section 5.5.3).

### 5.5.1 Employing Engaging Question Classifier

We propose to learn an engaging question classifier for (1) filtering out non-engaging questions generated by GPT-4 during the dataset generation process (Section 4.2), and (2) filtering out non-engaging questions produced by our model during inference (Section 5.2). This section examines the effect of employing this classifier.
**Dataset Generation.** To verify the effectiveness of filtering out questions from GPT-4 with the classifier, we train a T5-Base model to learn from an unfiltered dataset that contains all the questions produced by GPT-4 (Unfiltered Dataset). We compare the performance of this model to the T5-Base model trained on our proposed filtered dataset (Filtered Dataset) and report the human evaluation re-

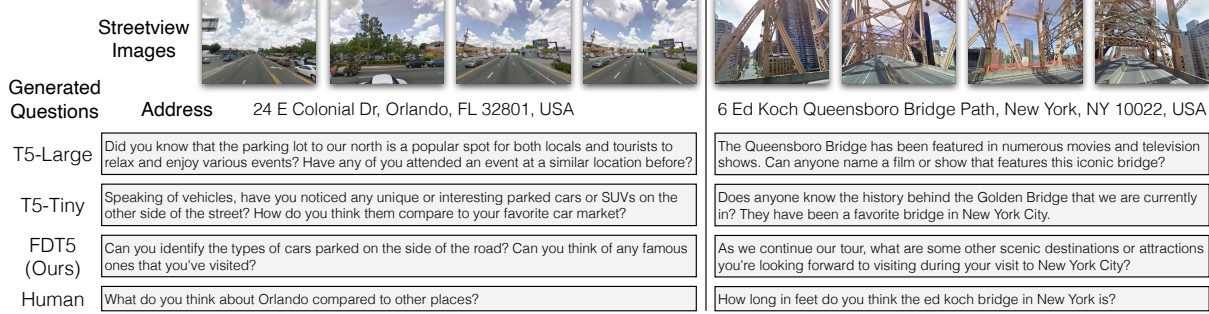

**Figure 4: Qualitative Results.** We present sampled generated questions from T5-Large, T5-Tiny, our proposed method FDT5, and human annotators, together with corresponding streetview images and addresses. With only 15M parameters, FDT5 can reliably generate engaging location-aware questions.

sults in Table 8. The results demonstrate that the model learning from the filtered dataset achieves better performance, justifying the efficacy of employing the classifier.

**Inference.** We propose to filter out non-engaging questions generated during inference, adopted in our method FDT5. We conduct human evaluations on filtered generation questions (Filtered Inference) and unfiltered questions (Unfiltered Inference), reported in Table 9. The results show that filtering non-engaging generated questions with the classifier can significantly improve the question quality on all the metrics. This justifies the effectiveness of employing the classifier during inference.

### 5.5.2 Incorporating GPS Coordinates

While Yeh et al. (2022) explored generating questions from a set of images, our work further incorporates addresses (reverse geocoded from GPS coordinates) into the question generation process. This section investigates the effect of employing such information. We compare the questions generated by GPT-4 with or without the address in the prompt and report the results in Table 10. The results show that incorporating the address leads to richer and more diverse questions, verifying the unique value of the proposed LocaVQG task.

| Criteria | w/o address | w/ address (ours) |
|---|---|---|
| Vocabulary Size ↑ | 450 | **525** |
| Average Question Length ↑ | 25.02 | **30.18** |
| Yngve Score ↑ | 3.531 | **3.693** |

Table 10: **Effect of Leveraging Street Address** The questions generated with street addresses are richer and more diverse.

### 5.5.3 Varying Dataset Sizes

We investigate the impact of varying dataset sizes with T5-Tiny and our proposed FDT5, and report the results n Table 11. FDT5 achieves better performance with fewer data points and performs comparably to T5-Tiny when dataset size increases. This indicates that our method is more data efficient.

| Model | #Samples | BLEU-4 | ROUGE-2 | BERTScore | BLEURT |
|---|---|---|---|---|---|
| | 0.7K | 0.2566 | 0.0366 | 0.5160 | -0.7666 |
| | 1.7K | 0.2629 | 0.0341 | 0.5139 | -0.7530 |
| T5-Tiny | 2.7k | 0.2604 | **0.0374** | 0.5164 | -0.7589 |
| | 3.7K | 0.2635 | 0.0371 | **0.5156** | 0.7419 |
| | 4.7K | **0.2639** | 0.0361 | 0.5145 | **-0.7398** |
| | 0.7k | 0.2565 | 0.0361 | 0.5201 | -0.7149 |
| | 1.7k | 0.2700 | 0.0402 | **0.5214** | -0.7245 |
| FDT5 (Ours) | 2.7k | 0.2675 | **0.0422** | 0.5211 | 0.7126 |
| | 3.7k | 0.2661 | 0.0393 | 0.5190 | **-0.7073** |
| | 4.7k | **0.2706** | 0.0386 | 0.5180 | -0.7256 |

Table 11: **Effect of Dataset Size.** From the results, FDT5 is more data efficient as it could achieve better performance with smaller sample size

### 5.6 Qualitative Results

As human evaluations can be subjective, we present qualitative results in Figure 4 for the readers to better understand the generated questions. The results show that FDT5 with only 15M parameters can reliably generate engaging location-aware questions.

## 6 Conclusion

In this work, we propose a novel task, location-aware visual question generation (LocaVQG), which aims to generate engaging questions from data relevant to a particular geographical location. Specifically, we represent location-aware information using four directional street view images and a GPS coordinate. To address this task, we introduce a dataset generation pipeline that leverages the recent advances of large language models (*i.e.*,

GPT-4) to generate diverse and sophisticated questions. To ensure the engagingness of the questions produced by GPT-4, we employ an engaging question classifier to filter out non-engaging questions. Our proposed dataset contains richer and various questions compared to existing datasets.

To learn from the proposed LocaVQG dataset with lightweight models, we present Filtered Distilled T5-Tiny (FDT5) method. We extensively evaluate the proposed method and the baselines with human evaluation and automatic evaluation metrics. Our proposed FDT5, with the fewest parameters, demonstrates superior performance on most metrics. We conduct extensive ablation studies to verify the effect of employing the filtering classifier, the effectiveness of incorporating GPS coordinates into the question generation process, and the impact of varying dataset sizes. We hope this work will encourage researchers to explore the LocaVQG task and its applications.

## Limitations

We discuss the limitations and how we can potentially address them in this section.

**Biases in AMT workers.** We notice that the AMT workers involved in human evaluation might be biased due to their demographic. This can potentially be addressed by ensuring the diversity of their background.

**Location-aware information.** Aiming to develop an in-car intelligent assistant, this work proposes representing location-aware information as a GPS coordinate and a set of images captured by on-car cameras. Incorporating more detailed information, such as local news and weather, can potentially lead to more diverse and engaging questions, and is left for future research.

**Address-aware LLMs.** Our proposed dataset generation pipeline heavily relies on GPT-4. This partially limits the generated questions to locations/addresses that are known by GPT-4 and therefore this pipeline might not produce coherent questions given locations that are less known by GPT-4. We can potentially address this by employing a more sophisticated external information retrieval system to extract information from a location

**Human evaluation setup.** While our motivation is to develop an in-car intelligent system that can engage a driver in a conversation to keep the driver awake, this work falls short from the following perspectives. First, our work solely focuses on generating a question without considering continuing a conversation. Second, we evaluate the generated questions with an AMT interface where the AMT workers read and evaluate the questions. However, in a driving scenario, interacting with a virtual assistant by reading a question is impractical. Hence, evaluating the generated questions by connecting to a text-to-speech system and requiring the annotators to rate the questions by listening to them would align better with the application.

**Distractingness of generated questions.** This work makes the very first attempt to develop an in-car visual question generation system that can ask engaging questions to initiate conversations with drivers. However, such engaging questions can potentially distract drivers and lead to dangerous situations in the worst case. To address such a concern, we encourage future works along this line to consider the "distractingness" of generated questions. In particular, developing evaluation metrics to determine if a question would distract a driver from the road and devising methods to produce engaging yet non-distracting questions are promising and interesting research directions.

## Ethics Statement

Since our proposed dataset generation pipeline involves collecting questions from GPT-4, the data inherits any biases of GPT-4. Moreover, our proposed method learns from the dataset, and therefore will also be biased. Therefore, inheriting biases can lead to generating inappropriate, sexist, racist questions or descriptions. Fortunately, addressing ethical issues has been an active research area (Liang et al., 2021; Baldini et al., 2021; Yan et al., 2023; Kasneci et al., 2023). We wish to incorporate the advances in the field to alleviate ethical concerns.

## Acknowledgement

This work is supported by the National Science and Technology Council of Taiwan under grants 111-2634-F-002-022- and by the Academia Sinica and Hong Hai Research Institute collaborative project 05T-1110523-1Q. Shao-Hua Sun was partially supported by the National Taiwan University and its Department of Electrical Engineering, Graduate Institute of Communication Engineering, and College of Electrical Engineering and Computer Science, and the Yushan Fellow Program by the Ministry of Education, Taiwan. We thank the online crowd workers for participating in the study.

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

## A    Additional Diversity Analyses on GPT-4 Generated Questions

We perform further diversity analyses on the questions provided in our LocaVQG dataset.

- **Question type analysis**: While Table 3 shows the top 15 most frequent trigrams of generated questions, we have performed an additional trigram analysis during the rebuttal period to examine the diversity of the generated questions. In particular, we followed Yeh et al. (2022) and identified 2437 question types among our 35K generated questions. This highlights the diversity of the generated questions.

- **Pairwise cosine similarity**: Inspired by Schwenk et al. (2022), which computes the average pairwise cosine similarity between each pair of generated questions encoded by a sentence transformer (multi-qa-MiniLM-L6-cos-v1 provided by HuggingFace) in a dataset, we have performed this evaluation on our generated dataset. We obtained an average cosine similarity of 0.1698, indicating that the generated questions are not highly correlated and therefore ensuring the diversity of our proposed dataset.

## B    Latency Analysis

Our goal is to develop lightweight models that can run on mobile devices. To examine the applicability of our proposed model FDT5 and the baselines, we measure and report the latency of MVQG-VL-T5, T5-Large, T5-Base, T5-Tiny, and our proposed FDT5 in Table 15. Each inference and post-filtering time is computed by averaging over 300 trials to reduce the variance.

| Latency (sec) | Loading Model | Inference | Post-Filtering |
|---|---|---|---|
| MVQG-VL-T5 | 7.09 | 6.38 | N/A |
| T5-Large | 12.79 | 10.04 | N/A |
| T5-Base | 10.34 | 5.9 | N/A |
| T5-Tiny | 3.89 | 2.02 | N/A |
| FDT5 | 4.25 | 2.27 | 3.92 |

Table 15: **Latency Testing of the trained models.**

The results show that FDT5 and T5-Tiny, with the same model architecture and the same number of parameters, enjoy a significantly reduced time for loading models and running inference. The post-filtering phase of FDT5 takes 3.92 seconds on average, indicating that the engaging question classifier requires FDT5 to perform 1.73 additional inference trials for each LocaVQG task on average. Note that this post-filtering phase can be shut down for latency-critical scenarios, and FDT5 without post-filtering still outperforms T5-Tiny in human evaluation, according to Table 6 and Table 9.

## C    Filtering Out Non-engaging Questions Generated by GPT-4 Using GPT-4

This work proposes to train an engaging question classifier to filter out non-engaging questions generated by GPT-4; alternatively, we can use GPT-4 to evaluate and filter out non-engaging questions that it generates. To investigate this possibility, we feed the questions generated by GPT-4 back into GPT-4 for scoring (*i.e.*, determining if each generated question is engaging or not).

Specifically, we provide GPT-4 with 10 questions generated by itself and asked it to determine if each question is engaging or not. The prompt and the response of GPT-4 are shown in Table 12. As a result, GPT-4 classifies all the questions it generated as engaging. This suggests that GPT-4 may not be useful to score its own generated questions, and motivates learning an external model to score the questions, such as our proposed engaging question classifier.

Note that this finding aligns with the statement, "we observe that GPT-4 assigns significantly higher scores to its own outputs compared to human ratings," stated in Dettmers et al. (2023).

## D    Effect of Incorporating Address Information

We are using the address as the main cue for GPT-4 to retrieve some information regarding those places. While it is possible to produce general, non-location-specific questions based on hand-crafted templates, we found that GPT-4 can certainly produce location-specific questions that require knowledge of specific locations, which can potentially be more engaging. Some examples are presented as follows.

- Did you know that Fort Duquesne Boulevard is named after the historic Fort Duquesne, which was a key location during the French and Indian War? Have any of you studied that period in history?

| Prompting GPT-4 for Self-Evaluation |
| --- |
| You will be given 10 questions. Please classify this question into engaging and non-engaging questions. |

1. Did you know that Pittsburgh is known as the "City of Bridges" due to its 446 bridges? Can you spot any nearby bridges from this location?
2. The tall buildings on the south side are a part of Pittsburgh's beautiful skyline. Can anyone name some of the prominent buildings that make up the skyline?
3. As we pass by these parked cars on the north side, can you guess the most popular car brand in Pittsburgh based on what you see?
4. Pittsburgh has a rich history of steel production, earning it the nickname "Steel City." Can you see any indications of this industrial past in the architecture of the buildings around us?
5. The traffic light at the empty intersection on our east might make you wonder about the city's traffic patterns. What do you think contributes to the current flow of traffic in this area?
6. Pittsburgh is known for its vibrant arts scene. Can anyone spot a nearby art gallery or theater as we drive along these city streets?
7. As we pass by the buildings on the west, let's talk about the city's diverse architectural styles. Can you identify different styles just by looking at their exteriors?
8. Pittsburgh is also famous for its rich sports culture, with successful teams in football, baseball, and hockey. Can you spot any sports fans or sports-themed establishments in our surroundings?
9. The city has a growing tech scene, with companies like Google and Uber opening offices here. Can you see any tech companies or startups as we drive through these streets?
10. Pittsburgh is known for its friendly locals, often called "yinzers" due to their unique accent. Have you encountered any friendly yinzers during your visit so far, and what has been your favorite interaction with them?

| GPT-4 Response |
| --- |

1. Engaging
2. Engaging
3. Engaging
4. Engaging
5. Engaging
6. Engaging
7. Engaging
8. Engaging
9. Engaging
10. Engaging

Table 12: **GPT-4 Self Evaluation Results.**

- The city of Pittsburgh is known for its numerous bridges. How many bridges do you think are in the city, and why do you think there are so many?

- The 59th Street Bridge, also known as the Ed Koch Queensboro Bridge, connects Manhattan to Queens. Can you identify any famous movies or TV shows that have featured this iconic bridge?

We cannot obtain these questions by simply replacing the city name in other questions, nor should these questions be asked at a different location.

Moreover, while generating a question with an address can increase the vocabulary size and average question length by simply inserting the address into the question, we still observe some questions that are generated based on the knowledge extracted by GPT-4 according to the address. We provide some example questions as follows.

- **Address**: 1250 Penn Ave, Pittsburgh, PA 15222, USA

  – **Generated question**: "As we drive along 1250 Penn Ave, are there any upcoming events, festivals, or celebrations in the area that you'd like to learn more about?"

  – **Observation**: GPT-4 knows this location has hosted several events in the past and therefore asks about upcoming events.

- **Address**: 333 Boulevard of the Allies, Pittsburgh, PA 15222, USA

  – **Generated question**: Did you know that the Boulevard of the Allies is named to honor the Allies of World War I? What do you think about the significance of this historical connection?

  – **Observation**: Based on knowing the history of the Boulevard of the Allies, GPT-4 asks about World War I.

# E Importance of Incorporating Visual Input and Learning to Generate Questions

Since it is possible to generate questions solely based on the fetched address, we aim to further analyze the effect of employing visual inputs to produce questions. Also, as discussed in Section D, we aim to quantitatively compare generating questions by our learned model and producing questions using general hand-crafted templates. To this end, we labeled 100 questions generated by FDT5 based on the following two criteria:

- The generated questions contain visual information (w/ vis) or not (w/o vis)

- The generated questions are based on some templated (templated) or a learned language model (learned)

Specifically, we went through each question and

- Determined if it contains visual information (e.g., describing surroundings). If so, this question is labeled as w/ vis; otherwise, it is labeled as w/o vis.

- Decided if it can be generated based on some templates (e.g., the city's name can be replaced with another city and the question still makes sense). If so, this question is labeled as templated; otherwise, it is labeled as learned.

Then, we analyze the engagement score and diversity of each group of questions. Regarding engagement scores, the questions containing visual information (w/ vis) achieve an average score of 4.007, slightly outperforming the questions without visual information (w/o vis) with an average score of 3.915, indicating that the visual-related questions may be more engaging. On the other hand, as the reviewer anticipated, the templated questions have a higher average score of 4.002 compared to learned questions (3.901).

Then, we analyzed the diversity of each group of questions and found that the learned questions are more diverse (with a pairwise Cosine similarity score of 0.3614), outperforming the templated questions with a pairwise Cosine similarity score of 0.3995.

In conclusion, we believe that generating engaging and diverse questions still requires incorporating visual inputs with a well-learned language model.

# F Implementation Details

## F.1 GPT-4 Parameters and Expenses

**Setup.** When generating our proposed dataset using GPT-4, we use the model "gpt4" listed in the OpenAI API, with 0.7 temperature and 0.1 presence penalty.

**Expenses.** On average, each request of our task uses up around 500 tokens, costing us around $0.001 USD. In total, generating the dataset and experimenting with it cost around $150 - 200$ USD.

## F.2 T5 and FDT5

**Input.** Similar to GPT-4, the T5 models and FDT5 take image captions as input. The text input to the T5 models and FDT5 is modified from the chat prompt provided to GPT-4. Specifically, we prepend "generate questions:" prefix to each input, resulting in the model input as: *generate questions: You are currently driving on [Street Address]. On your North, [Image Caption]. On your East, [Image Caption]. On your South, [Image Caption]. On your West, [Image Caption].*

**Implementation.** We adopt the basic pre-trained T5 models available on the Hugging Face platform.

**Training.** During the training, we use 5 questions for each LocaVQG task tuple. We train each model for 20 epochs with a learning rate of $10^{-4}$.

## F.3 VL-T5

**Input.** The input of VL-T5 contains the following prompt prefix, street address, visual embeddings, and visual semantic groundings.

- **Prompt prefix.** The prompt prefix is *generate question:*, which guides the model to generate questions with the instruction.

- **Street address.** The street address is the specific street address of the pictures that is verbalized, *e.g.*, *You are currently driving in Penn Avenue, Pittsburgh.*

- **Visual embeddings and visual semantic groundings.** We extract the visual embeddings from the whole image and the image regions with Faster-RCNN (Ren et al., 2015). Also, we adopted the grounded situation recognition (GSR) (Pratt et al., 2020) model to extract information on the sequence of images to understand the semantics. Prefix of the directions of the images is also added, *e.g.*, *North: [Visual embeddings]*

**Training.** During the training, we used the pre-trained baselines presented in (Yeh et al., 2022), specifically the adapter. We train the model for 30 epochs with a learning rate of $10^{-5}$. We also employ gradient accumulation steps of 4 and warm-up steps of 10.

### F.4 Engaging Question Classifier

The engaging question classifier is trained on the questions from the two datasets: SQuaD (20239 questions) and MVQG (31098 questions). The engaging question classifier is a BERT-based classifier with 110M parameters. We train the classifier to classify the questions sampled from SQuaD as non-engaging and those sampled from MVQG as engaging for 10 epochs. We use the ADAM optimizer with a learning rate of $10^{-5}$.

Some example questions are as follows:

- Why is that man playing billiards by himself? (Engaging)

- How did you celebrate your last birthday party? (Engaging)

- What document was signed in 1999? (Non-engaging)

- What did John Milton do for world literature? (Non-engaging)

We report the performance of the learned engaging question classifier on the training (train), validation (val), and testing (test) sets in Table 14. The accuracy evaluates if the classifier can correctly distinguish the questions in the MVQG dataset from those in the SQuaD dataset. The results show that the trained engaging question classifier can accurately distinguish the questions from the two datasets.

|  | train | valid | test |
|---|---|---|---|
| **Accuracy** | 99.9% | 98.9% | 99.0% |

Table 14: **Performance of the Learned Engaging Question Classifier.**

## G Amazon Mechanical Turk Details

### G.1 Human Evaluation Details

Section 5.3 conducts human evaluation on AMT to compare questions generated by all models, GPT-4, and humans. For each generated question, each worker is provided with the street address and 4 streetview images of this location. Then, the worker is required to rate the generated question according to 5 evaluation metrics: Engagement, Naturalness, Coherence, Common Sense, and Grounding. The descriptions of the metrics are as follows.

- **Engagement** This is an engaging question for this set of photos. You would want to answer or respond to this question

- **Naturalness** Given the pictures and location that you are in, it is natural to ask this question.

- **Coherence** This question asks something about the description and information that could be found in the image or relevant to the location.

- **Common Sense** This question provide enough Common Sense. The question asks something that makes sense according to our common sense.

- **Grounding** This question mentions the essential objects or information of the images, and it is mentioned in the right direction or talking about the location.

This AMT interface is shown in Figure 5

### G.2 Collecting Human Generated Questions

In order to compare GPT-4 to humans regarding the ability to produce engaging questions from location-aware information, we crowdsource human-generated questions from human annotators on AMT. We use three-stage questions to collect questions as follows.

- **Question 1.** Pick 1 (or more) pictures that you want to focus on, and write down the object (or event) that stands out the most for you.

- **Question 2.** Please describe the object (or event).

- **Question 3.** Please write a question based on it.

This AMT interface is shown in Figure 6. For each LocaVQG task tuple, we require three AMT workers to write a question, and then take the best question from the three questions. We present some questions produced by the workers in Table 14.

| |
|---|
| There is a big strong building around us. Do you want to try come and visit it? |
| Can you give me some tips to how to drive as like you? |
| what do you think this building is used for? |
| where are your coming from and which place you are going to visit? |
| Look at the building around us, what do you think about the building? I find it looking really sturdy |
| On the east side, you could see a cargo with few people around it. What do you think is inside the cargo? |
| Have you ever wondered what is beyond that colossal vertical gateway adorned with glass windows of elegance? I wonder if it could it be an art gallery or perhaps a stock exchange? It is Pittsburgh ya know, the Šteel Cityïhe possibilities are endless! |
| What do you think about Garland Avenue compared to other streets in Orlando? |

Table 14: **Human-Generated Questions.**

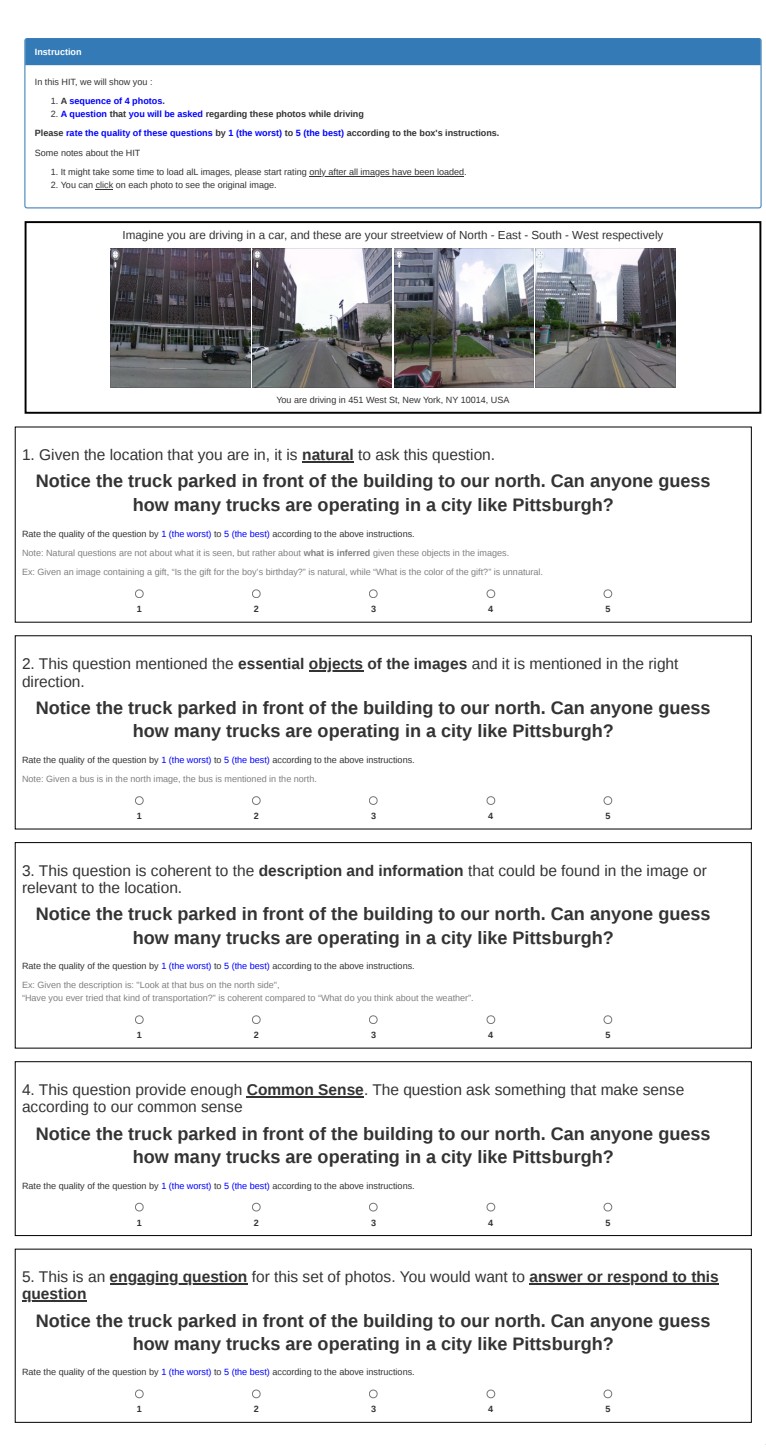

Figure 5: **Human Evaluation AMT Interface.**

**Instructions**

In this HIT, you will see a sequence of photos of your surrounding while driving (Streetview)

The first photo is your north view, second photo is your east view, third is your south view, and fourth is your west view

Please take a look at the photo sequence carefully and **answer three questions.**

You can submit the HIT after **30** seconds!

This is the requirement for the **third query**

1. Natural Question: Natural questions are not about what is seen, but rather about what is inferred given these objects.
   - **Prompt: given an image of a building with a large garage**
   - Good Question Example: What do you think is inside the garage?
   - Bad Question Example: What is the color of the building?
2. Engaging Question: Questions that sparks conversation, or pique other interests.
   - **Prompt: given an image with a beautiful riverside park. The images are taken from street in Pittsburgh**
   - **Use the Location information to make more engaging question!**
   - Good Question Example: What do you think about riverside park in Pittsburgh compared to other places?
   - Bad Question Example: What is the name of the river?
3. Non-generic and not event-centric: Generic questions can appear in most circumstances
   - Bad Question Example: What is that building? What's the weather? What color is that?

**Note: Please think and read the description of the box carefully before answering. If your answering time is too short, your answer may be rejected**

- It will take some time to load all images, please start answering only after all 4 images have been loaded.
- You can click on each photo to see the original image.

Please look at this 4 images of streetview images:

Note: From left to right, North - East - South - West

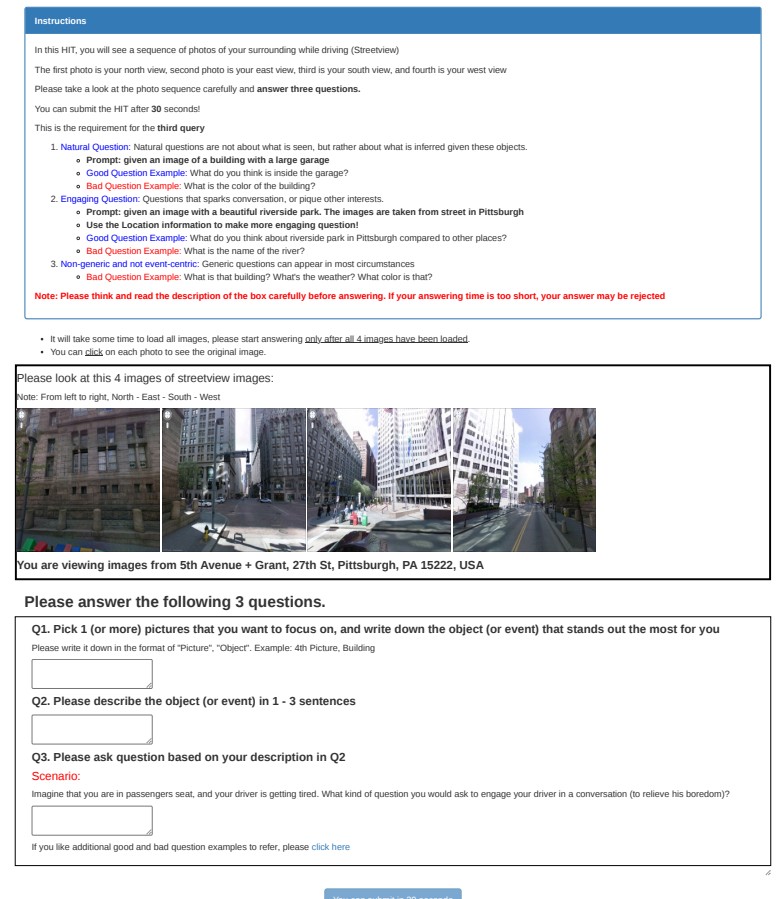

**You are viewing images from 5th Avenue + Grant, 27th St, Pittsburgh, PA 15222, USA**

### Please answer the following 3 questions.

**Q1. Pick 1 (or more) pictures that you want to focus on, and write down the object (or event) that stands out the most for you**

Please write it down in the format of "Picture", "Object". Example: 4th Picture, Building

**Q2. Please describe the object (or event) in 1 - 3 sentences**

**Q3. Please ask question based on your description in Q2**

Scenario:

Imagine that you are in passengers seat, and your driver is getting tired. What kind of question you would ask to engage your driver in a conversation (to relieve his boredom)?

If you like additional good and bad question examples to refer, please click here

You can submit in 29 seconds

Figure 6: **Location-aware Question Collection AMT Interface.**