# OpenReview forum: "Location-Aware Visual Question Generation with Lightweight Models"
_EMNLP/2023/Conference — EMNLP 2023 Main_

### Official Review · Reviewer_BPM2 · 2023-07-28

**Soundness:** 4

**Excitement:**

4: Strong: This paper deepens the understanding of some phenomenon or lowers the barriers to an existing research direction.

**Paper Topic And Main Contributions:**

This paper presents a new task, location-aware visual question generation (LocaVQG), along with the related dataset and a model (FDT5). The task aims to generate engaging and relevant questions based on the geographic coordinates of the user's location and the images of the user's surroundings. The dataset for the task is collected by employing GPT-4 to generate such questions given a prompt that includes location information and automatically produced captions of the images. The resulting questions are further filtered to keep only the engaging ones. The authors also present a lightweight model to produce such questions given the same input data. The model FDT5 is a pre-trained T5-Large model, fine-tuned based on the dataset of questions generated by GPT-4, followed again by "engagement" filtering.

**Questions For The Authors:**

A. Regarding the questions that urge the driver to pay attention to particular aspects of the surrounding area (e.g. the architectural style of the buildings). Couldn't they be considered dangerously distracting from the road conditions and traffic? Maybe the questions should be judged not only as "engaging" and "not engaging" but also as "distracting" and "not distracting". For example, a question about the local cuisine doesn't require the driver to take their eyes off the road and therefore is not as distracting as the "architectural style" question.

**Reasons To Accept:**

The proposed task is novel and interesting, with potentially important real-world applications. The presented dataset and method, which shows a strong performance, can serve as a benchmark and baseline for future research.

**Reasons To Reject:**

1. The paper is missing details of the engaging question classifier (dataset size and composition, model and training details).
2. The metrics used in the automatic evaluation (Section 5.4) rely on comparing the generated text to some "gold standard", ground truth text. In this case, it is not clear what is used as the ground truth. Questions generated by GPT-4 for the original dataset? But if some model generates a different question from what GPT-4 has generated previously, it does not mean at all that the new question is any worse. This section should be extended with a more detailed description of how the automatic evaluation was conducted and how the authors interpret its results given the concerns above.
3. At least some of the results reported in Table 10 seem to be trivial. Adding concrete address information to the prompt leads to named entities (location names) being generated in the questions, which trivially increases the vocabulary size and average question length (the latter by simply adding more input that questions are based on). These findings also do not necessarily relate to the quality of the resulting questions (how engaging/relevant they are, etc.).


UPD AFTER REBUTTAL: I trust the authors will include the explanations regarding these 3 points (just like they did in the rebuttal) in the revised version, so I'm raising the scores relying on that.

**Reproducibility:**

4: Could mostly reproduce the results, but there may be some variation because of sample variance or minor variations in their interpretation of the protocol or method.

**Reviewer Confidence:**

3: Pretty sure, but there's a chance I missed something. Although I have a good feel for this area in general, I did not carefully check the paper's details, e.g., the math, experimental design, or novelty.

**Typos Grammar Style And Presentation Improvements:**

Table 2 caption, lines 344, 382, 390: missing a full stop at the end.

434: "the two techniques employed in FDT5" -- unclear, better to explicitly state what these techniques are

References for GPT-4, BERTScore, ROUGE-2 should be given at their first mentions in the paper (currently they appear much later than this). References for BERT, GPT-3 should also be given when they are mentioned in the paper.

---

> ### Author Rebuttal · Authors · 2023-08-26
>
> We sincerely thank the reviewer for the thorough and constructive comments. Please find the response to your questions below.
>
> ### Responses to Questions
>
> > The paper is missing details of the engaging question classifier (dataset size and composition, model and training details).
>
> We thank the reviewer for pointing this out. We provide the missing details as follows and will incorporate them into the revision.
>
> The engaging question classifier is trained on the questions from the two datasets: SQuaD [1] (20239 questions) and MVQG [2] (31098 questions). The engaging question classifier is a BERT-based classifier with 110 M parameters. We train the classifier to classify the questions sampled from SQuaD as non-engaging and the questions sampled from MVQG as engaging for 10 epochs. We use the ADAM optimizer [3] with a learning rate of 1e-5. Some example questions are as below.
> - “Why is that man playing billiards by himself?” (engaging)
> - “How did you celebrate your last birthday party?” (engaging)
> - “What document was signed in 1999?” (non-engaging)
> - “What did John Milton do for world literature?” (non-engaging)
>
> > The metrics used in the automatic evaluation (Section 5.4) rely on comparing the generated text to some "gold standard", ground truth text. In this case, it is not clear what is used as the ground truth. Questions generated by GPT-4 for the original dataset? But if some model generates a different question from what GPT-4 has generated previously, it does not mean at all that the new question is any worse. This section should be extended with a more detailed description of how the automatic evaluation was conducted and how the authors interpret its results given the concerns above.
>
> Given a LocaVQG task tuple sampled from our dataset, the automatic evaluation compares a generated question with a sampled GPT-4 generated question from this tuple. More details on the automatic evaluation are described as follows.
> - **BLEU** and **ROUGE** compare two questions based on the exact wording; therefore the scores may not exactly reflect the quality of generated questions.
> - **BLEURT** and **BERTScore** use an ML model to compare two questions, which may better reflect the similarity in semantics.
>
> We agree with the reviewer that the automatic evaluation may not precisely reflect the quality of the generated questions. Therefore, our work uses human evaluation as our main evaluation metric. Yet, **the automatic evaluation can still indicate how well the trained models fit the dataset distribution**. As a result, we believe it is still informative to provide the automatic evaluation results. We will revise the paper to incorporate this information.
>
> > At least some of the results reported in Table 10 seem to be trivial. Adding concrete address information to the prompt leads to named entities (location names) being generated in the questions, which trivially increases the vocabulary size and average question length (the latter by simply adding more input that questions are based on). These findings also do not necessarily relate to the quality of the resulting questions (how engaging/relevant they are, etc.).
>
> **We provide the experimental results presented in Table 10 because we aimed to explicitly differentiate our work from MVQG [2].** Specifically, MVQG generates questions from a set of images, while our work generates questions from a set of images and a GPS coordinate. Therefore, we believe it is still informative to provide such a study for readers familiar with MVQG and its extensions, even if the results may seem trivial.
>
> Moreover, while generating a question with an address can increase the vocabulary size and average question length by simply inserting the address into the question, **we still observe some questions that are generated based on the knowledge extracted by GPT-4 according to the address**. We provide some example questions as follows.
> - **Address**: 1250 Penn Ave, Pittsburgh, PA 15222, USA
>     - **Generated question**: “As we drive along 1250 Penn Ave, are there any upcoming events, festivals, or celebrations in the area that you'd like to learn more about?”
>     - **Observation**: GPT-4 knows this location has hosted several events in the past and therefore asks about upcoming events.
> - **Address**: 333 Boulevard of the Allies, Pittsburgh, PA 15222, USA
>     - **Generated question**: Did you know that the Boulevard of the Allies is named to honor the Allies of World War I? What do you think about the significance of this historical connection?
>     - **Observation**: Based on knowing the history of the Boulevard of the Allies, GPT-4 asks about World War I.
>
> >  Regarding the questions that urge the driver to pay attention to particular aspects of the surrounding area (e.g. the architectural style of the buildings). Couldn't they be considered dangerously distracting from the road conditions and traffic? Maybe the questions should be judged not only as "engaging" and "not engaging" but also as "distracting" and "not distracting". For example, a question about the local cuisine doesn't require the driver to take their eyes off the road and therefore is not as distracting as the "architectural style" question.
>
> We thank the reviewer for pointing this out. Our work makes the very first attempt to develop an in-car visual question generation system aiming to alleviate driver’s fatigue by asking engaging questions to initiate conversations. During our preliminary study, we found that generating location-aware, engaging questions is particularly challenging, and therefore this work aims to tackle this problem.
>
> We agree with the reviewer that future works along this line should consider the “distractingness” of generated questions. In particular, developing evaluation metrics to determine if a question would distract a driver and devising methods that can produce engaging yet non-distracting questions are promising and interesting research directions. We will revise the paper and incorporate this into our limitations and future work.
>
> > Table 2 caption, lines 344, 382, 390: missing a full stop at the end.
>
> > 434: "the two techniques employed in FDT5" -- unclear, better to explicitly state what these techniques are
>
> We thank the reviewer for the suggestions. We will thoroughly revise the paper to fix them.
>
> > References for GPT-4, BERTScore, ROUGE-2 should be given at their first mentions in the paper (currently they appear much later than this). References for BERT, GPT-3 should also be given when they are mentioned in the paper.
>
> We thank the reviewer for this suggestion. We will revise the paper to properly provide the references.
>
> ### References
>
> - [1] Rajpurkar et al. “SQuAD: 100,000+ questions for machine comprehension of text” EMNLP 2016
> - [2] Yeh et al. “Multi-VQG: Generating engaging questions for multiple images” EMNLP 2022
> - [3] Kingma et al. “Adam: A Method for Stochastic Optimization” ICLR 2015
>
> ### Conclusion
>
> We are incredibly grateful to the reviewer for the detailed and constructive review. We believe our responses address the concerns raised by the reviewer. Please kindly let us know if there are any further concerns or missing experimental results that potentially prevent you from accepting this submission. We would be more than happy to address them if time allows. Thank you very much for all your detailed feedback and the time you put into helping us to improve our submission.

---

### Official Review · Reviewer_Gg8Z · 2023-08-04

**Typos Grammar Style And Presentation Improvements:** Line 406 -- 3.98 instead of 3.88? Or …
**Soundness:** 4

**Excitement:**

4: Strong: This paper deepens the understanding of some phenomenon or lowers the barriers to an existing research direction.

**Paper Topic And Main Contributions:**

The paper proposes a new task called location-aware visual question generation (LocaVQG), which is basically visual question generation conditioned on a given location. They motivate this by citing how engaging drivers in conversation/QA can reduce fatigue and help the drivers stay attentive.

**Dataset** -- The input consists of a location (e.g. "PPG Place, Third Avenue, Pittsburgh, PA") and 4 corresponding images (north, south, east, west cameras). The images were taken from the Google Streetview Dataset and the location is reverse-geocoded. Their dataset contains 3,759 such examples.

**Methodology** -- To generate engaging questions, they do the following: 1) caption each image to convert to caption. 2) generate a prompt given the location + 4 captions (north, south, east, west images) (e.g. "You are driving in [...]. On your north, [...]. On your south, [...]. Please generate 10 engaging questions.") 3) This prompt is then fed into GPT-4 to generate questions. 4) The questions are fed into an engaging question classifier to classify which questions are engaging.

**Experiments and Evaluation** -- They train a T5-Tiny model (15.6M) by distilling from a T5-large model and from GPT-4. Evaluation is done through MTurk human evaluation (asking humans to evaluate Engagement, Coherence, etc. on a 1-5 scale), as well as through automatic evaluation metrics (BLEU, ROUGE, BERTScore, etc.) They show that their lightweight T5-Tiny model performs better than other lightweight models. Lastly, there is a thorough ablation study on the various parts of the pipeline/dataset.

**Questions For The Authors:**

- How are the images selected from Google Streetview?
- How do you select which question to use as the ground truth for fine-tuning? Is this randomly selected?
- Is there a way for you to measure the diversity of the questions you're generating? (see above point for more details)

**Reasons To Accept:**

They propose a new task (location-based VQG), and I think these new tasks are nice for the community especially now that people are excited about the capabilities of these new language models. In terms of the applications track, I also think this is a well-done application-based paper because it's both well-motivated in the real world (useful for drivers), as well as interesting from a research standpoint.

They do a distillation onto a smaller model, which is getting quite popular these days with large models. Usually I would be wary of simply distilling large models as a research idea, but in this case, I think it's justified because the authors are using a very tiny model.

Thorough analysis of the dataset. Evaluations make sense. Nice to see that they have human evaluations since automatic evaluations wouldn't really mean much for this task. The paper is generally clear and well-written.

**Reasons To Reject:**

I'm not sure how valuable this task will be for the NLP community. I agree that it's an important and challenging task, but I'm not sure that it's a benchmark that a lot of people will be working on. I'm also not entirely convinced that having the visual part of the pipeline actually helps. I wouldn't be surprised if just using the locations can provide enough information for the model to generate something informative. For instance, a lot of the "engaging questions" in Table 1 don't refer to the images at all and can be generated solely by using the questions.

I'm not very convinced with the engaging question classifier. A lot of the "engaging" questions in Table 1 seem very generic (e.g. about food, architecture, etc.) that you can just swap out the name of the city and it's going to be a perfectly valid Barnum question for any location you want. I think perhaps a test for diversity could be helpful here.

I'm also not entirely convinced with the results in Table 6. T5-Large, T5-Base, and T5-Tiny all perform around the same, and the gap between the GPT-4 (4.04) and T5-Tiny (3.85) is very small, which makes me somewhat question whether the human evaluations are actually suitable for this task or if humans are just poor at scoring these questions.

I would have wanted to see a discussion on speed/latency. This usually isn't an issue for most papers, but this paper claims one of its main points to be that it can work on edge devices such as mobile phones, which opens up the whole can of worms on latency/inference time. For instance, they mention that they filter out poor questions until the model produces a question that's classified as engaging. I'm wondering how many times the model generated a rejected question before it generates one that is accepted.

**Reproducibility:**

4: Could mostly reproduce the results, but there may be some variation because of sample variance or minor variations in their interpretation of the protocol or method.

**Reviewer Confidence:**

3: Pretty sure, but there's a chance I missed something. Although I have a good feel for this area in general, I did not carefully check the paper's details, e.g., the math, experimental design, or novelty.

---

> ### Author Rebuttal · Authors · 2023-08-26
>
> We sincerely thank the reviewer for the thorough and constructive comments. Please find the response to your questions below.
>
> ### Responses to Questions
>
> > I'm not sure how valuable this task will be for the NLP community. I agree that it's an important and challenging task, but I'm not sure that it's a benchmark that a lot of people will be working on.
>
> We appreciate the reviewer for recognizing the importance and the challengingness of the task introduced in our work. We aim to leverage the rapid advancement in NLP to revolutionize how we interact with technology and build systems that enhance human ability. To this end, our work makes the initial attempt to develop an in-car visual question generation system that can ask engaging location-aware questions to keep drivers awake. **We believe our work will inspire further innovation in various visual question generation tasks and a wide range of systems that promote road safety.**
>
> > I'm also not entirely convinced that having the visual part of the pipeline actually helps. I wouldn't be surprised if just using the locations can provide enough information for the model to generate something informative. For instance, a lot of the "engaging questions" in Table 1 don't refer to the images at all and can be generated solely by using the questions.
>
> We thank the reviewer for raising this concern. We acknowledge that the questions presented in Table 1 are not particularly relevant to the visual part (i.e., streetview images) of LocaVQG task tuples. This is because we only aimed to compare engaging and unengaging questions in Table 1. **We provide several GPT-4 generated questions relevant to visual inputs as follows.**
> - "As we look towards the south, can you guess the purpose of this brick building with cars parked in front? Perhaps an office building, a restaurant, or something else?"
> - "What do you think the history of this western city street might be? Pittsburgh has a rich history, and every street has a unique story to tell."
> - "What types of events or festivals might take place in this park throughout the year?"
>
> **The above questions refer to a building, a park, or a street that could be seen from the images, which are based on the visual part of the LocaVQG task tuples.** We will revise Table 1 to include these questions to avoid confusion. We will also include a discussion on the importance of employing visual cues for visual and language tasks [1, 2] in the revision.
>
> > I'm not very convinced with the engaging question classifier. A lot of the "engaging" questions in Table 1 seem very generic (e.g. about food, architecture, etc.) that you can just swap out the name of the city and it's going to be a perfectly valid Barnum question for any location you want.
>
> We agree with the reviewer that one can potentially generate some questions with some hand-crafted templates. However, designing such templates requires expertise and may be time-consuming. Instead, our work proposes to minimize human effort by prompting GPT-4. While, as pointed out by the reviewer, some questions in Table 1 are generic, **GPT-4 can certainly produce location-specific questions**. We provide some example questions sampled from the proposed dataset below.
> - “Did you know that Fort Duquesne Boulevard is named after the historic Fort Duquesne, which was a key location during the French and Indian War? Have any of you studied that period in history?”
> - “The city of Pittsburgh is known for its numerous bridges. How many bridges do you think are in the city, and why do you think there are so many?”
> - “The 59th Street Bridge, also known as the Ed Koch Queensboro Bridge, connects Manhattan to Queens. Can you identify any famous movies or TV shows that have featured this iconic bridge?”
> - “The plaza to the south is often used for various events and activities throughout the year. Have any of you attended or heard about any events that take place at PPG Place?”
>
> **We cannot obtain these questions by simply replacing the city name in other questions, nor should these questions be asked at a different location.** This highlights GPT-4’s ability to generate location-specific and diverse questions. We will revise the paper to clarify this and incorporate these questions into Table 1 to avoid confusion.
>
> > Is there a way for you to measure the diversity of the questions you're generating?
>
> We agree with the reviewer that examining the diversity of the generated questions is indeed crucial. **We have analyzed the diversity of the generated questions from the following perspectives.**
> - **Comparison to an existing VQG dataset (MVQG)**: Therefore, we have compared our proposed dataset to an existing visual question generation dataset (MVQG) [2], and the results are presented in Table 5 (Question Quality Comparison with MVQG). The comparisons are based on vocabulary size, average sentence length, syntactic complexity (Yngve score normalized by the sentence length), percentage of abstract terms, and average term depth. **The results show that the proposed dataset contains significantly more diverse and sophisticated questions compared to MVQG.**
> - **Question type analysis**: While Table 3 shows the top 15 most frequent trigrams of generated questions, we have performed an additional trigram analysis during the rebuttal period to examine the diversity of the generated questions. In particular, **we followed [2] and identified 2437 question types among our 35K generated questions**. This highlights the diversity of the generated questions.
> - **Pairwise cosine similarity**: Inspired by [3], which computes the average pairwise cosine similarity between each pair of generated questions encoded by a sentence transformer (multi-qa-MiniLM-L6-cos-v1 [4]) in a dataset, we have performed this evaluation on our generated dataset. We obtained an average cosine similarity of 0.1698, indicating that **the generated questions are not highly correlated and therefore ensuring the diversity of our proposed dataset.**
>
> We will incorporate the additional analysis into our paper. While we have strived for comprehensive diversity analysis during the rebuttal period, if the reviewer has other specific diversity metrics in mind, we will be happy to use them to evaluate our proposed dataset and report them in the revision.
>
> > I'm also not entirely convinced with the results in Table 6. T5-Large, T5-Base, and T5-Tiny all perform around the same
>
> We agree with the reviewer. The performance of the three T5 models with different sizes (T5-Large: 3.87, T5-Base: 3.84, T5-Tiny: 3.85) is very similar. Therefore, this motivates us to develop a method based on T5-Tiny. We would like to emphasize that our proposed method **FDT5 outperforms all the T5 models** with an overall score of 3.98.
>
> > … the gap between the GPT-4 (4.04) and T5-Tiny (3.85) is very small
>
> To answer the reviewer’s question, we have performed a t-test statistical analysis on the performance of T5-Tiny and GPT-4. As a result, we obtained a p-value of 0.1459. This indicates **there is statistical significance in the difference in the performance of T5-Tiny (3.85) and GPT-4 (4.04)**. This motivates us to develop a method that performs more comparably to GPT-4 and has significantly fewer parameters.
>
> > I would have wanted to see a discussion on speed/latency. This usually isn't an issue for most papers, but this paper claims one of its main points to be that it can work on edge devices such as mobile phones, which opens up the whole can of worms on latency/inference time. For instance, they mention that they filter out poor questions until the model produces a question that's classified as engaging. I'm wondering how many times the model generated a rejected question before it generates one that is accepted.
>
> We thank the reviewer for raising this concern about speed and latency. We also believe that incorporating this information into our paper will strengthen the paper.  **We have measured and reported the latency of MVQG-VL-T5, T5-Large, T5-Base, T5-Tiny, and our proposed FDT5 below.** Each inference and post-filtering time is computed by averaging over 300 trials to reduce the variance.
>
> | Latency | Loading Model (sec) | Inference (sec) | Post-filtering (sec) |
> | :---: | :---: | :---: | :---: |
> | MVQG-VL-T5 | 7.09 | 6.38 | N/A |
> | T5-Large | 12.79 | 10.04 | N/A |
> | T5-Base |  10.34 | 5.9 | N/A |
> | T5-Tiny |  3.89 | 2.02 | N/A |
> | FDT5 |  4.25 | 2.27 | 3.92 |
>
> The results show that FDT5 and T5-Tiny, with the same model architecture and the same number of parameters, enjoy a significantly reduced time for loading models and running inference. The post-filtering phase of FDT5 takes 3.92 seconds on average, indicating that the engaging question classifier requires FDT5 to perform 1.73 additional inference trials for each LocaVQG task on average. Note that this post-filtering phase can be shut down for latency-critical scenarios, and FDT5 without post-filtering still outperforms T5-Tiny in human evaluation, according to Table 6 and Table 9.
>
> We will incorporate the results and the discussion above in the revision.
>
> > How are the images selected from Google Streetview?
>
> The Google Streetview dataset contains locations that are very close to each other. Therefore, we subsampled the dataset by picking one coordinate from every three adjacent coordinates. Each coordinate contains five directional images: North, East, South, West, and Up (upward view). We discarded the upward view and used the remaining four directional images to form the images in a LocaVQG tuple.
>
> > How do you select which question to use as the ground truth for fine-tuning? Is this randomly selected?
>
> Yes. At each training iteration, we first randomly sample a batch of LocaVQG tuples from the dataset. Then, we randomly sample a question from each LocaVQG tuple as the ground truth question for training.
>
> > Line 406 -- 3.98 instead of 3.88? Or is the table wrong?
>
> We thank the reviewer for pointing out this error. It should be 3.98. We will revise the paper and fix it.
>
> ### References
>
> - [1] Huang et al., “Summaries as Captions: Generating Figure Captions for Scientific Documents with Automated Text Summarization” INLG 2023
> - [2] Yeh et al., “Multi-VQG: Generating Engaging Questions for Multiple Images” EMNLP 2023
> - [3] Schwenk et al. “A-OKVQA: A Benchmark for Visual Question Answering using World Knowledge” arXiv 2022
> - [4] [multi-qa-MiniLM-L6-cos-v1 on Hugging Face](https://huggingface.co/sentence-transformers/multi-qa-MiniLM-L6-cos-v1)
>
> ### Conclusion
>
> We are incredibly grateful to the reviewer for the detailed and constructive review. We believe our responses address the concerns raised by the reviewer. Please kindly let us know if there are any further concerns or missing experimental results that potentially prevent you from accepting this submission. We would be more than happy to address them if time allows. Thank you very much for all your detailed feedback and the time you put into helping us to improve our submission.

---

### Official Review · Reviewer_BRvQ · 2023-08-12

**Paper Topic And Main Contributions:** 1. This work introduces a novel task …
**Soundness:** 4

**Excitement:**

4: Strong: This paper deepens the understanding of some phenomenon or lowers the barriers to an existing research direction.

**Questions For The Authors:**

Is it necessary to have an externally trained engaging scorer? Wouldn't it be feasible to simply input the generated questions back into GPT-4 for scoring?

**Reasons To Accept:**

The acceptance of this article could potentially facilitate the application of LLMs within specific vertical domains. This work demonstrates a comprehensive pipeline for the application of LLMs in a specific vertical field, which involves annotating data by prompting LLMs, filtering data for desired feature (engagement here), and training more time-efficient models.

**Reasons To Reject:**

1. The thoughts of prompting LLMs to collect data has limited novelty.
2. The effectiveness of the engaging question classifier in the study seems to be uncertain. There is a lack of evaluation regarding the performance of the trained engaging question classifier. Also, There is a lack of evaluation regarding the engagement of the generated question.

**Reproducibility:**

4: Could mostly reproduce the results, but there may be some variation because of sample variance or minor variations in their interpretation of the protocol or method.

**Reviewer Confidence:**

3: Pretty sure, but there's a chance I missed something. Although I have a good feel for this area in general, I did not carefully check the paper's details, e.g., the math, experimental design, or novelty.

---

> ### Author Rebuttal · Authors · 2023-08-26
>
> We sincerely thank the reviewer for the thorough and constructive comments. Please find the response to your questions below.
>
> ### Responses to Questions
>
> > The thoughts of prompting LLMs to collect data has limited novelty.
>
> As pointed out by the reviewer, prompting LLMs to collect data has been recently explored in [1, 2]. While [1] focuses on using LLM to generate instruction data for instruction tuning, [2] aims to leverage LLM as a teacher in the distillation process. To the best of our knowledge, **our work is the first to prompt LLMs to generate data for visual question generation (VQG) tasks**. Specifically, this work proposes a dataset generation pipeline that leverages GPT-4 for location-aware VQG. The contributions of the proposed dataset generation pipeline include converting images to text with an image captioning model, reverse geocoding to obtain an address, designing effective system and chat prompts, and utilizing an engaging classifier to filter generated questions. We believe these contributions are novel and valuable.
>
> > The effectiveness of the engaging question classifier in the study seems to be uncertain. ...  Also, There is a lack of evaluation regarding the engagement of the generated question.
>
> We propose to utilize the proposed engaging question classifier to filter out unengaging questions generated by GPT-4 when generating the dataset and filter out unengaging questions produced by our proposed question generation method FDT5. **We extensively evaluate and discuss the effectiveness of this proposed engaging question classifier in Section 5.5.1 with the results presented in Table 8 and Table 9.** We provide an overview of the results as follows.
> - **Dataset generation pipeline**: The results presented in Table 8 show that the model learning from the filtered dataset achieves better performance in terms of both the Engagement and Overall scores in human evaluation, compared to the model learning from the unfiltered dataset. This justifies the efficacy of employing the proposed engaging question classifier in the dataset generation pipeline.
> - **Filtering FDT5 generated questions**: The results presented in Table 9 compare the questions produced by our proposed method FDT5 with or without filtering out the generated questions deemed unengaging by the proposed engaging classifier. With the proposed filtering procedure, the generated questions achieve higher Engagement and Overall scores in human evaluation. This verifies the effectiveness of employing the proposed engaging question classifier during inference.
>
> Please refer to section 5.1.1 for detailed discussions on the effectiveness of the engaging question classifier.
>
> > There is a lack of evaluation regarding the performance of the trained engaging question classifier.
>
> We thank the reviewer for the suggestion. We report **the performance of the learned engaging question classifier on the training (train), validation (val), and testing (test) sets** as follows. The accuracies evaluate if the classifier can correctly distinguish the questions in the MVQG dataset from the questions in the SQuaD dataset. We will revise the paper and incorporate this information into the paper.
>
> | | train | val | test |
> | --- | --- | --- | --- |
> | Accuracy | 99.9% | 98.9% | 99.0% |
>
> > Is it necessary to have an externally trained engaging scorer? Wouldn't it be feasible to simply input the generated questions back into GPT-4 for scoring?
>
> We thank the reviewer for the suggestion. We did not use GPT-4 to determine whether the questions generated by itself are engaging or not because recent studies have shown that **LLMs tend to be biased and prefer their own generated answers**. This is evident from [3], which states that “we observe that GPT-4 assigns significantly higher scores to its own outputs compared to human ratings."
>
> To empirically verify the above statement, we have conducted an additional experiment that inputs the generated questions back into GPT-4 for scoring (i.e., decide if each generated question is engaging or not). We provided GPT-4 with ten questions generated by itself and asked it to determine if each question is engaging or not. The prompt and the GPT-4 output are shown as follows. As a result, **GPT-4 classifies all the questions it generated as engaging**. This suggests that GPT-4 may not be useful to score its own generated questions, and motivates learning an external model to score the questions, such as our proposed engaging question classifier.
>
> We thank the reviewer for this suggestion. We will incorporate this additional experiment and discussion into our revision to strengthen our paper.
>
> Prompt to GPT-4
> ```
> You will be given 10 questions. Please classify this question into engaging and non-engaging questions.
>
> 1. Did you know that Pittsburgh is known as the \"City of Bridges\" due to its 446 bridges? Can you spot any nearby bridges from this location?
> 2. The tall buildings on the south side are a part of Pittsburgh's beautiful skyline. Can anyone name some of the prominent buildings that make up the skyline?
> 3. As we pass by these parked cars on the north side, can you guess the most popular car brand in Pittsburgh based on what you see?
> 4. Pittsburgh has a rich history of steel production, earning it the nickname \"Steel City.\" Can you see any indications of this industrial past in the architecture of the buildings around us?
> 5. The traffic light at the empty intersection on our east might make you wonder about the city's traffic patterns. What do you think contributes to the current flow of traffic in this area?
> 6. Pittsburgh is known for its vibrant arts scene. Can anyone spot a nearby art gallery or theater as we drive along these city streets?
> 7. As we pass by the buildings on the west, let's talk about the city's diverse architectural styles. Can you identify different styles just by looking at their exteriors?
> 8. Pittsburgh is also famous for its rich sports culture, with successful teams in football, baseball, and hockey. Can you spot any sports fans or sports-themed establishments in our surroundings?
> 9. The city has a growing tech scene, with companies like Google and Uber opening offices here. Can you see any tech companies or startups as we drive through these streets?
> 10. Pittsburgh is known for its friendly locals, often called \"yinzers\" due to their unique accent. Have you encountered any friendly yinzers during your visit so far, and what has been your favorite interaction with them?
> ```
>
> GPT-4’s Response
> ```
> 1. Engaging
> 2. Engaging
> 3. Engaging
> 4. Engaging
> 5. Engaging
> 6. Engaging
> 7. Engaging
> 8. Engaging
> 9. Engaging
> 10. Engaging
> ```
>
> ### References
>
> - [1] Wang et al. “Self-Instruct: Aligning Language Models with Self-Generated Instructions” ACL 2023
> - [2] Hsieh et al. “Distilling Step-by-Step! Outperforming Larger Language Models with Less Training Data and Smaller Model Sizes” ACL 2023
> - [3] Dettmers et al. “QLoRA: Efficient Finetuning of Quantized LLMs” arXiv 2023
>
> ### Conclusion
>
> We are incredibly grateful to the reviewer for the detailed and constructive review. We believe our responses address the concerns raised by the reviewer. Please kindly let us know if there are any further concerns or missing experimental results that potentially prevent you from accepting this submission. We would be more than happy to address them if time allows. Thank you very much for all your detailed feedback and the time you put into helping us to improve our submission.

---

### Meta-Review · Area_Chair_CuaX · 2023-09-16

**Recommendation:** 5

**Metareview:**

The paper proposes the task of location based visual question generation. They use GPT-4 to produce diverse engaging questions. And then share studies on both the data generation as well as solving the problem.

The reviewers find the paper exciting and sound after the rebuttal. As one reviewer indicates, there are flaws but it's nonetheless interesting. The authors should include several points raised by the reviewers so as to improve the paper: clarifications on the prompting and filtering, metrics and eval changes suggested by reviewers, and question classifier details to name a few.

---

### Decision · Program_Chairs · 2023-10-07

**Decision:**

Accept-Main

**Comment:**

The paper proposes the task of location based visual question generation. They use GPT-4 to produce diverse engaging questions. And then share studies on both the data generation as well as solving the problem.

The reviewers find the paper exciting and sound after the rebuttal. As one reviewer indicates, there are flaws but it's nonetheless interesting. The authors should include several points raised by the reviewers so as to improve the paper: clarifications on the prompting and filtering, metrics and eval changes suggested by reviewers, and question classifier details to name a few.